# Senescence: Pathogenic Driver in Chronic Obstructive Pulmonary Disease

**DOI:** 10.3390/medicina58060817

**Published:** 2022-06-17

**Authors:** Melissa Rivas, Gayatri Gupta, Louis Costanzo, Huma Ahmed, Anne E. Wyman, Patrick Geraghty

**Affiliations:** 1Department of Medicine, State University of New York Downstate Medical Centre, Brooklyn, NY 11203, USA; melissa.rivas@downstate.edu (M.R.); louis.costanzo@downstate.edu (L.C.); huma.ahmed@downstate.edu (H.A.); anne.wyman@downstate.edu (A.E.W.); 2Section of Pulmonary, Critical Care and Sleep Medicine, Yale University School of Medicine, New Haven, CT 06520, USA; gayatri.gupta@yale.edu

**Keywords:** chronic obstructive pulmonary disease, senescence, cigarette smoke, aging

## Abstract

Chronic obstructive pulmonary disease (COPD) is recognized as a disease of accelerated lung aging. Over the past two decades, mounting evidence suggests an accumulation of senescent cells within the lungs of patients with COPD that contributes to dysregulated tissue repair and the secretion of multiple inflammatory proteins, termed the senescence-associated secretory phenotype (SASP). Cellular senescence in COPD is linked to telomere dysfunction, DNA damage, and oxidative stress. This review gives an overview of the mechanistic contributions and pathologic consequences of cellular senescence in COPD and discusses potential therapeutic approaches targeting senescence-associated signaling in COPD.

## 1. Introduction

Chronic obstructive pulmonary disease (COPD) is a heterogeneous condition with multiple phenotypes that gives rise to progressive and irreversible airflow limitation [1]. It is a highly prevalent disease and a major cause of morbidity and mortality worldwide [2]. Exposure to smoke from cigarettes or biomass fuel is the most significant risk factor for the development of COPD [3,4]. These exogenous insults trigger injury, inflammation, and structural remodeling of the airways and lung parenchyma [5,6]. However, only a minority of smokers develop COPD, suggesting alternative factors such as viral infection or genetic differences that may contribute to its development [7]. Genetic predisposition may partly determine the host’s susceptibility and response to environmental stressors. Genome-wide association studies have identified multiple genetic risk loci for COPD and its subsets [8]. COPD is an aging-related disease, with the incidence rate rising with increasing age [9]. During the normal aging process, pulmonary function begins to decline as a consequence of reduced elastic recoil of the lung and reduced compliance of the chest wall [10]. The reduced elasticity largely stems from homogeneous, nondestructive dilation of the alveolar spaces, which decreases the surface tension of the alveoli [11,12]. Reduced compliance of the chest wall can be attributed to musculoskeletal changes that alter the shape of the thorax and increase the forces needed to move the chest [13]. Elevated levels of reactive oxygen species (ROS), low-grade inflammation, shortened telomeres, and an increased number of senescent cells are observed in the aging lung [14].

Although elevated ROS are an essential part of natural aging, the appropriate levels are exceeded in COPD [6]. This is a consequence of the high concentrations of inhaled oxidants and the endogenous release of ROS by inflammatory cells, epithelial cells, and endothelial cells, thereby creating an oxidant/antioxidant imbalance [15]. The lungs accumulate a greater number of senescent lung fibroblasts, as well as epithelial and endothelial cells when compared to the healthy aging lung [16,17,18,19]. The elevated frequency of senescent cells may be due to ROS and oxidative stress. Senescent cells further propagate inflammation due to the adoption of a hyper-secretory phenotype, known as the senescence-associated secretory phenotype (SASP) [20,21]. Accumulation of senescent cells can be attributed to immunosenescence, a process that attenuates both innate and adaptive immunity [22]. Here we will give an overview of molecular mechanisms and mediators contributing to the development of cellular senescence in COPD, discuss the contributions of senescent cells to disease pathology, and review the development of therapies targeting senescent cells that hold promise in treating COPD. Improved understanding of the role of senescence in aging and COPD should pave the way for more effective treatments.

## 2. Overview of Cellular Senescence

### 2.1. Definition

Cellular senescence describes a state of irreversible growth arrest characterized by morphologic changes and secretion of multiple extracellular factors (such as cytokines, growth factors, and proteases) with autocrine and paracrine effects defined as the SASP. Although markers specific to the senescent state or universal to all senescent cells are not yet identified, most senescent cells express senescence-associated β-galactosidase (SA-βgal) activity, a result of increased lysosomal mass in senescent cells, and show an absence of proliferative markers and increased expression of cell cycle inhibitors, tumor suppressors, and DNA damage markers [23]. See Table 1 for the different forms of senescence, SASPs associated with each type of senescence, some of the pathways altered in these scenarios, and whether the SASPs and the pathways are implicated in playing a role in early emphysema development.

### 2.2. Cellular Pathways

Cellular senescence was first described nearly sixty years ago in fibroblasts that have lost their proliferative potential due to a permanent state of cell cycle arrest [61]. The limited proliferation of cells in vitro is now recognized as a particular type of senescence resulting from telomere loss [62,63]. Senescence may occur during embryonic development or in response to multiple stressors such as DNA damage, oncogenic mutations, oxidative stress, mitochondrial dysfunction, and autophagy inhibition [21,23,62]. These stimuli activate multiple signaling pathways, including those mediated by MKK3/MKK6, RAS, MYC, PI3K, and TGFβ, which lead to the downstream activation of cell cycle inhibitors and the tumor suppressor retinoblastoma protein (RB) [62]. These pathways interact to promote senescence through de-phosphorylation and activation of RB, which arrests cell proliferation by preventing the transcription of genes involved in the S phase of the cell cycle [64]. Cyclin-dependent kinase (CDK) inhibitors, which maintain RB in an unphosphorylated state, include the INK4 members p15, p16, p18, and p19 and the CIP/KIP members p21, p27, and p57. Activation of the tumor suppressor p53 by the ataxia-telangiectasia mutated kinase (ATM)/Rad3-related kinase (ATR) and Chk1/Chk2 kinases following genomic damage promotes senescence through subsequent p21 activation and RB dephosphorylation [64,65].

### 2.3. Markers of Senescence

Defining senescence in cell cultures and tissues generally requires a combination of markers and functional assays, which include the CDK inhibitors p16 and p21, measurements of telomere length, biochemical assays that measure senescence-associated products such as SA-βgal and lipofuscin, proliferation studies, and markers of DNA damage such as phosphorylated histone H2AX (γ-H2AX) [21,62]. Finding reliable markers of senescence in COPD is limited by several factors. First, senescent cells exhibit significant heterogeneity with distinct proteomic and gene expression profiles depending on cell type and stressors [21,62,66]. Second, senescence biomarkers, such as the pro-inflammatory factors characteristic of the SASP, are often non-specific, occurring in other cellular contexts. Third, multiple molecular mediators of senescence likely contribute to disease pathology, requiring a collection of markers to establish a link between senescence and relevant disease outcomes. For example, p16 knock-out (KO) mice exposed to chronic cigarette smoke (CS) do not have decreases in lung inflammation or airspace enlargement compared to controls, suggesting that pathways other than those regulated by p16 contribute to CS-induced cellular senescence in COPD and emphysema [38]. Despite these complexities, multiple studies have shown associations between senescent markers and pathologic processes in COPD, as described in the following sections.

### 2.4. Immunosenescence

Senescent cells are cleared by a mixture of macrophages, natural killer cells, cytotoxic T cells, B cells, neutrophils, mast cells, eosinophils, and dendritic cells. They make use of pattern recognition receptors and release SASP factors that attract immune cells [67]. However, these immune cells are subject to aging, which causes impaired phagocytosis, chemotaxis, and bactericidal activity. Furthermore, antigen presentation and pattern recognition signaling become less effective [22]. The pool of naïve B and T lymphocytes contracts while their corresponding memory cells undergo clonal expansion [68]. Levels of CD4 + CD28null and CD8 + CD28null cells increase, which leads to inflammation but compromises immune surveillance [68]. Many COPD patients display dysregulated immunity overall, with CD8+CD28null cells demonstrating reduced HDAC2 expression and corticosteroid resistance [69,70].

## 3. Overview of Evidence for Cellular Senescence in COPD

### 3.1. Cellular Senescence Increased in COPD

Several groups have shown an increase in senescent cells in patients with COPD compared to age-matched smokers or healthy non-smoking controls. Higher percentages of type II pneumocytes [24,27,71], endothelial cells [71,72,73], and pulmonary artery smooth muscle cells [74] with positive staining for p16 or p21 were observed in lung tissue sections from patients with COPD or emphysema compared to smokers [27,71,72,73] or non-smoking controls [27,71]. Compared to control smokers, endothelial [44,73] and smooth muscle [44,74] cells from patients with COPD showed increased SA-βgal staining and lower population doubling levels (PDLs). Increased p16 or p21 expression and increased percentages of SA-βgal-positive cells were also observed in fibroblasts [16], small airway epithelial cells (SAECs) [28], and endothelial progenitor cells [29] from patients with COPD compared to age-matched smokers [16] or non-smokers [28,29]. Significant correlations between markers for cellular senescence such as p16, p21, or population doubling levels in tissues and clinical parameters such as measures of airflow obstruction [71] lend additional support to the contribution of senescence to the development of COPD.

### 3.2. Contribution of CS to Cellular Senescence

Most studies investigating the effects of CS on senescence were performed on submerged cultured cells with CS extract (CSE), which can have different properties from mainstream CS and thereby different biological outcomes [75]. In in vitro, senescence is reduced early, but repeated exposure to CSE is required to induce an irreversible senescent state [76]. Studies have demonstrated that CSE induces cellular senescence associated with increased SA-βgal activity, cell morphology changes, or irreversible growth arrest through activation of the ATM-p53-p21 and p16-Rb pathways in multiple cell types, including SAECs and alveolar epithelial cells [24,25,26] and fibroblasts [25,26,28,76,77]. Similar to CSE in cell cultures, acute and chronic CS exposure increase senescence in mouse lung tissues as shown by increased SA-βgal activity, increased p16, p21, or p53 levels, and lipofuscin accumulation [24,26,35,36,37,38].

CSE and CS exposure may contribute to cellular senescence in cell cultures and murine lung tissues by inducing telomere damage or shortening [25,28,78], mitochondrial dysfunction [26,78], and DNA damage [37,38,79]. Both p16 and p21 were investigated in mouse models as mechanisms linking CS-induced cellular senescence to relevant outcomes in COPD, such as increased inflammatory responses, cellular apoptosis and proliferation, declines in lung function, and emphysema development. Deletion of p21 in mice attenuates CS-induced DNA damage, inflammatory responses, and oxidative stress [37,79] and promotes pulmonary epithelial cell proliferation [80], suggesting a protective effect of senescence-associated p21 inhibition in vivo. Increased p21 expression in alveolar macrophages from smokers is associated with a reduction in apoptosis induced by oxidative stress [30]. The deficiency of p16 in mice reduced SASP and inflammatory cytokine expression, increased type 2 pneumocyte proliferation, and protected against emphysema development in response to CS exposure [27]. Another study found that p16 deficiency protected against inflammatory cellular influx in response to acute CS exposure but did not attenuate declines in lung function or airspace enlargement in chronic CS-exposed mice [38]. Differences in outcome between these studies may be due to differences in p16 KO models used (p16-deficient mice in the latter study died prematurely due to spontaneous tumor formation in lymphoid organs) and suggest the need for further studies to delineate mechanisms regulated by p16 that contribute to pathogenic processes in COPD.

A recent study utilizing RNA sequencing approaches in CSE exposure and senescent conditions in airway epithelial cells demonstrated that CS and senescence conditions induce common signaling responses, including genes that regulate ROS, proteasome degradation, and NF-κB signaling [47]. Voic and colleagues report 243 common gene expression changes in epithelial cells when exposed to CSE or induced senescence [47]. Several of these genes are already reported to play significant roles in COPD pathogenesis, such as MMP-1 [81] and S100A9 [42], suggesting shared changes induced by CSE and senescence in the pathogenesis of COPD.

## 4. Mechanisms Contributing to Senescence in COPD

### 4.1. Telomere Dysfunction

Replicative senescence occurs once a cell has reached its Hayflick limit, the maximum number of times a cell is capable of dividing. Telomeres, tandem repeats of TTAGGG arranged in a T-loop structure at the ends of chromosomes, progressively shorten after each cell division until a critical length is reached and telomeres become uncapped [82]. Several studies have demonstrated that telomere attrition occurs in COPD. Decreased telomere lengths are observed in peripheral leukocytes [83,84], and endothelial cells [71,73] of COPD patients compared to non-smoking [71] or smoking [73,83,84] control subjects. However, a study by Müller, et al. demonstrated that senescent lung fibroblasts from patients with emphysema failed to display a reduction in telomere length [16]. Tsuji, et al. found a significant negative correlation between telomere length and p21 or p16 levels in type II alveolar and endothelial cells, showing an association between telomere length and senescence [71]. Thus, it was proposed that replicative senescence was at work here, perhaps as a consequence of the continuous alveolar turnover and regeneration that occurs in emphysema [71]. Telomere dysfunction can also induce senescence independently of telomere shortening. Birch, et al. found an increase in telomere-associated DNA damage foci (TAF) in SAECs from patients with COPD who had normal telomere lengths for their sex and age [28]. TAF co-localized with p16 in epithelial cells in lung tissue samples from patients with COPD, suggesting a role for TAF in inducing senescence. TAF were thought to have resulted from the telomeres’ susceptibility to oxidative stress [28]. This is a type of stress-induced premature senescence that is telomere-independent and is elicited by various factors that cause cellular stress, including CS. Houben, et al. showed lower levels of superoxide dismutase (SOD) in leukocytes from patients with COPD compared to controls and a positive correlation between telomere lengths and SOD [83], suggesting that oxidative stress may also contribute to telomere shortening in patients with COPD. Telomerase mutations, which lead to short telomeres and altered telomerase activity, were also shown to be a risk factor for severe or early-onset emphysema in patients who smoke [28,78,85]. Finally, maintaining sufficient telomere capping is another important means of preventing telomere dysfunction by protecting the chromosomes against deterioration or end-to-end fusion. CS can cause a reduction of the telomere capping protein (TPP1) that could augment cellular senescence in COPD/emphysema [25]. The E3 ubiquitin ligase FBW7 could be regulating this TPP1 response [85]. See Figure 1 for possible mechanisms to senescence in COPD.

### 4.2. DNA Damage

Phosphorylated histone H2AX (γH2AX) is a sensitive indicator of double-stranded DNA breaks and appears as distinct nuclear foci by immunofluorescent microscopy [72,86,87,88,89]. Once phosphorylated, γH2AX can recruit DNA repair proteins such as phosphorylated ATM/ATR substrates and phosphorylated 53BP1 to the damage site, and, if the DNA is unable to be repaired, a cell fate process is initiated [72]. Paschalaki, et al. observed an increased number of blood outgrowth endothelial cells (BOEC), endothelial progenitor cells with proliferative and angiogenic potential, containing γH2AX and 53BP1 nuclear foci in smokers and patients with COPD compared to healthy nonsmokers, which were age-independent [29]. Both γH2AX and 53BP1 expression correlated strongly with SA-βgal activity and smoking pack-years but not with age, suggesting a causative link between DNA damage and senescence which may be due to CS exposure [29]. Birch, et al. showed increased γH2AX foci colocalizing with telomeres in lung tissues and SAECs cultures from patients with COPD compared to controls, in aged compared to young mice, and in MRC-5 fibroblasts and primary SAECs exposed to CSE in vitro [28]. Similarly, Aoshiba, et al. demonstrated that alveolar type I and II cells and endothelial cells of COPD patients had a larger number of γH2AX foci than asymptomatic smokers and non-smokers. The greater number of foci observed in COPD patients was linked to apoptosis, senescence, or inflammation [72]. Increased 53BP1 foci were also observed in bronchial epithelial (Club) cells from mice with short telomeres, which, compared to wild-type animals, develop greater emphysema after CS exposure [80].

### 4.3. Oxidative Stress

The exact mechanism of oxidative stress-induced aging is still unknown. However, tissue damage in COPD from CSE-induced oxidative stress likely results from mitochondrial dysfunction and an imbalance between the production of ROS and the expression or activity of molecules involved in antioxidant responses [90]. Cigarette smoke contributes to cellular senescence in COPD by inducing mitochondrial dysfunction with increased ROS, DNA damage, and impaired mitophagy mediated by the PINK1-PARK2 pathway [26,39,40]. Specifically, CS exposure caused perinuclear accumulation of damaged mitochondria in human lung fibroblasts and SAECs due to reduced Parkin translocation, a family of proteins that function as ubiquitin E3 protein-ligases [51], to damaged mitochondria and cytoplasmic p53 accumulation [26]. Mitochondria-targeted antioxidant restored impaired mitophagy, decreased mitochondrial mass accumulation, and delayed cellular senescence in Parkin-overexpressing cells [26]. CSE stimulation reduced Miro1 and Pink1 levels in primary human epithelial cells to regulate mitophagy and mitochondria dysfunction [39]. Therefore, oxidative stress induced by CS exposure impacts mitochondrial function, which influences senescent responses. Oxidative stress induced by CS may also induce senescence by accelerating telomere dysfunction and activating the SASP (through IL-6 and IL-8 secretion) [28]. Additional evidence pertaining to oxidative stress is described in further detail in Section 5.1, Section 5.4, Section 5.8, Section 5.9 and Section 5.10 below.

## 5. Molecular Mediators of Senescence in COPD

### 5.1. Sirtuins

Several sirtuins (SIRTs), a group of class III deacetylases, are believed to have protective effects against COPD progression [91]. SIRT1 [31,32] and SIRT6 [31,33] levels are decreased in lung tissues, airway epithelial cells, and blood outgrowth endothelial cells from patients with COPD and smokers compared to NS controls [29] and in response to CSE or oxidative stress in a monocyte-macrophage cell line and human bronchial epithelial cells [32,33,92]. Recent evidence suggests that CSE-induced suppression of SIRT1 and SIRT6 levels or activity could lead to exaggerated senescence, an effect mediated through the upregulation of specific miRNAs in response to oxidative stress [33,93,94]. Levels of miR-34a are induced by oxidative stress through PI3K signaling and are increased in lung tissues and primary epithelial cells of patients with COPD compared to control subjects [31]. Inhibition of miR-34 in primary epithelial cells from patients with COPD increases SIRT1 and suppresses p16 and p21 levels, linking miR-34-mediated SIRT1 loss in COPD to cellular senescence. Oxidative stress induced by hydrogen peroxide stimulation also suppresses SIRT1 and SIRT6 mRNA and protein levels directly in bronchial epithelial cells [31]. MiR-34a has also been shown to induce senescence in lung fibroblasts [93] and negatively regulates apoptotic cell clearance (efferocytosis) in human and murine alveolar macrophages in part through SIRT1 suppression [94]. A recent study demonstrated that inhibition of miR-570-3p, a microRNA increased in COPD lung tissues and peripheral blood mononuclear cells, reverses cellular senescence by restoring the expression of SIRT1 [92]. Oxidative stress upregulates miR-570-3p expression through p38 MAP kinase-c-Jun signaling and miR-570-3p inhibition restores cellular growth and prevents SASP release in SAECs from COPD subjects [92,94]. Oxidative stress may also promote senescence by activating the DNA damage response (DDR), which negatively regulates SIRT1 levels [29]. Blood outgrowth endothelial cells from smokers and COPD patients have increased DNA double-strand breaks and senescence compared to nonsmokers. This senescent state is reversed with the SIRT1 activator, resveratrol [29]. Similarly, SIRT1 activation or overexpression protects against CSE-induced telomere DNA damage in lung fibroblasts [25,33]. SIRT6 inhibition by CSE contributes to senescence in HBEC through activation of IGF-Akt-mTOR signaling and insufficient autophagocytic removal of damaged cellular components [33].

### 5.2. MicroRNA (miRNA)

Other miRNAs, in addition to miR-34a and miR-570-3p described in Section 4.3, could influence senescence in COPD. Shen, et al. found that miR-200b was downregulated in a pulmonary emphysema mouse model and overexpressing miR-200b in mouse lung epithelial (MLE) cells attenuated CSE-induced cellular senescence, implicating miR-200b as a negative modulator of senescence in COPD [95]. Mechanistically, miRNA200b may protect against senescence and inflammation in MLE cells by downregulating the expression of zinc finger E-box binding homeobox 2 (ZEB2), which is a transcription factor consisting of many functional domains that interact with transcriptional co-effectors implicated in the attenuation of the inflammatory response pathway in pulmonary emphysema [95,96].

### 5.3. Klotho

Klotho, an antiaging gene, encodes a membrane-bound protein that may promote epithelial cell viability and protect against emphysema by regulating cigarette smoke-induced oxidative stress and cellular senescence [97]. Klotho overexpression in human bronchial epithelial (BEAS-2B) cells decreased ROS, increased p21 levels, and decreased cytotoxicity in response to CSE [97]. Cigarette smoke decreases klotho expression in primary airway epithelial cells, and klotho-deficient mice develop emphysema and increased airway inflammation [98].

### 5.4. Lamin B1

Alterations of lamin proteins, which maintain nuclear structural integrity [99] and regulate cell cycle progression, DNA replication, and gene silencing [100], are implicated in disorders of accelerated aging [101]. Loss of lamin B1 is associated with senescence [99,102]. Freund, et al. found that the decline of lamin B1 in senescent human and murine fibroblasts precedes the onset of morphological changes, SA-βgal activity, and SASP and occurs upon activation of either the p53-p21 pathway or p16-Rb regardless of the stimulus used to induce senescence. Saito, et al. reported reduced lamin B1 levels in airway epithelial cells from patients with COPD compared to smokers or non-smokers, in mice exposed to CS, and in HBECs treated with CSE. Lamin B1 silencing in normal primary HBECs induced senescence, which was enhanced by CSE treatment, as shown by increased SA-βgal, phosphorylated histone H2AX, p16, and p21 levels [78]. Furthermore, the expression of lamin B1 in HBECs was found to correlate directly with pulmonary function [78]. Reduced lamin B1 may be a promising marker for detecting senescence COPD and used to indicate disease severity.

### 5.5. Mammalian Target of Rapamycin (mTOR)

Houssaini, et al. showed activation of the mTOR signaling pathway in lung tissues and cultured endothelial and smooth muscle cells from patients with COPD compared to age- and sex-matched control smokers [44]. Lung expression of p16 correlated positively with p-Akt, p-GSK3, p-S6K, and p–4E-BP1 protein levels. Inhibition of mTOR by rapamycin increased cell PDLs and decreased percentages of SA-βgal-positive cells in patients with COPD and controls. Activation of mTOR signaling in mice through constitutive or conditional deletion of the tuberous sclerosis complex heterodimer TSC1, a negative mTORC1 regulator, in smooth muscle or endothelial cells induced senescence, emphysema, and pulmonary hypertension. Saito, et al. showed that CS-induced mTOR activation, mitochondria accumulation, and cellular senescence are mediated by lamin B reduction and downstream inhibition of DEPTOR, an mTOR regulator, in primary HBECs, murine lung airway epithelial cells, and COPD lungs [78]. Therefore, mTOR signaling could influence senescence in COPD pathogenesis.

### 5.6. Werner’s Syndrome Protein

Nyunoya, et al. demonstrated that CSE-induced senescence in fibroblasts was accompanied by a decrease in Werner’s syndrome protein (encoded by the *WRN* gene) [77]. The WRN protein interacts with proteins involved in telomere maintenance, DNA replication, and DNA repair [103]. This is of importance in the context of accelerated aging in COPD, as loss-of-function mutations of the *WRN* gene in Werner’s syndrome causes premature aging [104]. Loss of *WRN* expression in fibroblasts results in increased susceptibility to CS-induced cellular senescence and cell migration impairment [77]. The antioxidant treatment enhances WRN levels and reduces CSE-induced senescence [77].

### 5.7. Plasma Membrane Proteins

Caveolin-1, the structural protein component of caveolae, has been shown to protect against emphysema through the regulation of cellular senescence. Volonte, et al. showed that smoke-induced pulmonary emphysema and senescence were reduced in caveolin-1-deficient mice. Caveolin-mediated oxidative stress-induced senescence in lung fibroblasts through sequestration of PP2A and Mdm2, which led to downstream p53 and p21 activation [105]. CS also inhibits PP2A responses, which contributes to COPD pathogenesis [106], further linking caveolin-1 signaling to cellular senescence and emphysema.

### 5.8. Lipids

Prostaglandin E2 (PGE2), a pro-inflammatory lipid synthesized from arachidonic acid by cyclooxygenase, is produced by various resident cells of the airways, including epithelial cells, fibroblasts, and alveolar macrophages. In COPD, the cells overproduce PGE2, sometimes in amounts high enough to be detectable in exhaled breath [107]. Martien, et al. demonstrated that upregulation of COX2 and PGE2 occurs in senescent human lung fibroblasts [108]. They posited that COX2′s contribution to senescence is the generation of its own ROS species that can form DNA adducts. COX2′s overall impact on senescence, however, is mediated by PGE2 via PGE2 receptors [108]. Dagouassat, et al. showed that PGE2 secreted by senescent COPD fibroblasts induces cellular senescence and inflammation in neighboring fibroblasts in an autocrine and paracrine manner [109]. Cellular senescence induced by PGE2 and COX2 signaling also occurs in response to elevated 27-hydroxycholesterol (27-OHC) levels in the airways of patients with COPD [110]. Finally, insulin-like growth factor binding proteins (IGFBP) are elevated during exacerbations of COPD [111], and IGFBP-3 and -7 are linked to senescence-associated emphysema [16,27].

### 5.9. Creatinine Kinase

Creatinine kinase (CK) catalyzes the reversible phosphorylation of creatinine by ATP, allowing energy to be stored in the form of phosphocreatine (PCr), a fundamental process for maintaining the energy homeostasis of cells [112]. Dysregulation of the CK/PCr pathway is implicated in hypoxic and inflammatory disorders [113]. Energy status is considered an important determinant of senescence [112]. Activated 5′-adenosine monophosphate-activated protein kinase (AMPK)induced by increased AMP:ATP and ADP:ATP ratios during energy stress can induce senescence by directly phosphorylating p53 or by degrading the mRNAs of p16 and p21 inhibitors [114]. CK is subject to oxidation during ROS exposure which can impair enzymatic activity [112,115,116]. It is postulated that CS can downregulate CK and inactivate it by oxidation. Hara, et al. found that brain-type CK (CKB) levels and activity were markedly reduced in HBECs of smokers with COPD and that suppression of CKB led to induction of senescence [112].

### 5.10. Alpha-Antitrypsin Deficiency (AATD)

Telomere length is reported to be better preserved in peripheral blood cells in AATD patients with COPD than in non-deficient patients but is independent of changes in lung function both in subjects with AATD and in COPD controls [117]. Accelerated telomere attrition is observed in children and teenagers with AATD [118]. Therefore, senescence could influence the pathogenesis of AATD-related emphysema, but further studies are required.

## 6. Pathologic Consequences of Cellular Senescence in COPD

### 6.1. Dysregulated Inflammation

Kumar, et al. coined the term “COPD-associated secretory phenotype” (CASP) to refer to the inflammatory mediators that are increased in COPD and provided a comparison of CASP and SASP factors [119]. In summary, some of the factors that are upregulated in both CASP and SASP include the interleukins IL-1α, IL-1β, IL-6, IL-8, IL-13; chemokines GRO-α, GRO-β, GRO γ, MCP-2, MIP-1α, MIP- 3α; proteasesMMP-1, MMP-3, MMP-10, MMP-12, MMP-13, MMP-14; growth factors EGF, bFGF, VEGF, angiogenin, IGBP; and nitric oxide, ROS, and extracellular matrix proteins [119]. Their large degree of overlap supports the notion that they are strongly linked and reinforces the theory that senescence, along with SASP, is a major contributor to the inflammation that defines COPD.

Typically, cells undergoing senescence chemoattract immune cells, resulting in clearance of these senescent cells by immune cells such as NK cells and macrophages [120,121]. However, senescent cells in diseased tissues can also impede innate and adaptive immune responses [47,122]. As previously mentioned, senescent cells accumulate in tissues during aging [123,124] and could influence several pathological features observed in COPD, such as inflammation-associated tissue damage and remodeling. For comprehensive reviews of the influence of senescence on inflammatory responses, see Langhi Prata, et al. [67] and Vincente, et al. [125]. It is difficult to determine whether inflammation observed in COPD is primarily due to senescence, as many other contributing factors within the disease may contribute. However, the presence of enhanced senescent cell frequency in the lungs does contribute to a modified immune response that may influence several aspects of COPD pathogenesis.

Senescent cells secrete several factors that can influence inflammation responses, such as GM-CSF, GROα (and γ), MCP-1 to 4, IL-6, IL-8, IL-1β, MIP-1α (MIP-3α), MMPs (1, 3, 9, and 12), RANTES, RARRES2, TIMPs, and TGF-β [126,127]. Moreover, senescent cells secrete microvesicles, exosomes, microRNAs, other non-coding RNAs, mitochondrial DNA fragments, prostanoids, ceramides, bradykines, protein aggregates, and additional factors that could exacerbate inflammation [128,129,130,131]. The majority of these secreted cellular components are observed at elevated levels in age-related diseases, including COPD. Activation of either p21 or p16 is associated with the secretion of these potential inflammatory factors [132]. Conversely, chronic inflammation stresses cells and can lead to the spread of the senescent phenotype. This is observed in mice deficient for the anti-inflammatory genes *Il-10* [123] and *Nfκb1* [133], with elevated levels of senescent cells in both animal phenotypes. TNFα promotes senescence by inducing ROS and activating the JAK/STAT signaling pathway [134]. Oxidative stress-mediated inflammation is observed in COPD, most notably with loss of NRF2 expression resulting in suppressed antioxidant production, elevated ROS, and enhanced inflammation [135,136].

To fully understand the influence of senescent cells on the pathogenesis of COPD, we must first examine inflammatory-associated senescent cell signaling that could contribute to the disease process. The influence of age on inflammation is extensively studied, with aged mice exhibiting diminished anti-inflammatory potential in bone marrow-derived mesenchymal stromal cells [137]. Alternatively, transcriptome studies in multiple organs demonstrated that aging is characterized by the up-regulation of multiple genes that encode inflammatory mediators [138]. A recent systematic study of epigenomic and transcriptomic changes across tissues during aging in mice revealed the up-regulation of immune system response pathways, including the interferon response [139]. However, minimal transcriptome data is available for inflammation in senescent lung cells. Morrow, et al. demonstrated enrichment of COPD-relevant lung tissue B cell gene expression in peripheral blood, with CD28 expression altered in COPD [140]. CD28 is needed for effective primary T-cell expansion and activation of regulatory T-cells (Treg cells), and its loss could play a role in senescence and inflammation in COPD [141]. CD8/CD28(null) cells are increased in both current- and ex-smoker COPD subjects and these cells express more IFN-γ, OX40, 4-1BB, CTLA4, granzyme, and perforin [142]. Equally, mice exposed to CS have increased CD8/CD28(null) T cells in their airways [142], which could contribute to elevated inflammation. The same research group demonstrated that these CD28 null cells are senescent and exhibit elevated inflammation and enhanced glucocorticoid resistance [143]. Importantly, transcriptional noise increases with aging, possibly due to deregulated epigenetic control [144], which may contribute to the lack of extensive senescent transcriptional data in COPD. In the same single-cell sequencing study, a pro-inflammatory signature is observed in aging lungs, with upregulation of *Il6*, *Il1b*, *Tnf*, and *Ifnγ* and downregulation of *Pparg* and *Il10* [144]. New approaches to characterize senescence with transcriptome profiles will further enhance our knowledge of senescence in COPD [145].

### 6.2. Role of Senescence during Development and Disease

Although the word senescence means “to grow old” and senescent cells accumulate in aged tissues, senescence is not synonymous with aging. Senescence occurs during embryonic development and physiologically in adult cells such as megakaryocytes and placental syncytiotrophoblasts [62]. Developmentally programmed senescence is driven by the TGFβ/SMAD, PI3K/FOXO, and ERK signaling pathways and contributes to morphogenesis through the elimination of transient embryonic structures such as interdigital webs [146]. Senescent cells formed during normal development produce a SASP that can activate the immune system and apoptosis to induce their own clearance [62]. In adult tissues, reactivation of senescence-induced pathways in response to external stressors may have beneficial or harmful effects depending on the cellular trigger, cell type, or SASP [62]. Senescence has shown protective roles in some diseases and detrimental ones in others. The concept of senescence as a mechanism for tissue repair, remodeling, and regeneration that may be impaired during aging and particular disease states is particularly relevant to COPD given the risks of advanced age and exposure to damaging environmental stimuli that predispose to its development.

### 6.3. Impaired Tissue Regeneration

Overexpression of p16 contributes to replicative failure within many regenerative cell types [147,148]. Conversely, p16 downregulation ameliorates age-associated functional and proliferative impairments in stem and progenitor cells [149], suggesting that senescence contributes to the decline of tissue regeneration. In mice, short telomeres limit epithelial cell recovery after CS exposure, whereas p21 deficiency promotes increased alveolar and Clara cell proliferation in terminal bronchioles [80]. Similarly, Nyunoya and colleagues report that acute exposure to CS inhibits normal fibroblast proliferation required for lung repair, but chronic CS exposures trigger an irreversible state of senescence in cells that could contribute to the impaired tissue regeneration observed in COPD [76]. Extended exposure to CSE can induce two different fibroblast phenotypes: a senescent and a non-senescent phenotype, with the non-senescent cells exhibiting enhanced profibrotic signaling [150]. These non-senescent cells may contribute to fibrotic lesions in COPD, while the senescent cells contribute to emphysema development. In smokers and COPD patients, reduced angiogenic ability and increased apoptosis are observed in endothelial progenitor cells, which show increased senescence and DNA damage [29]. Recent evidence suggests that basal progenitor cells, which are important for airway epithelial differentiation, exhibit a reduced regenerative capacity in COPD [151,152], but this was deemed to be independent of senescence [153].

### 6.4. Dysregulation of Apoptosis and Cellular Proliferation

Elevated numbers of apoptotic alveolar epithelial [154] and endothelial [155] cells are found in the lungs of patients with COPD. Apoptosis-positive type II alveolar epithelial cells are observed in the alveolar walls of patients with pulmonary emphysema [156]. The apoptotic index is significantly higher in emphysematous lungs compared to controls (*p* ≤ 0.01), particularly in AATD emphysema [157]. Emphysema may result from different rates of cellular proliferation and apoptosis and the insufficient proliferative capacity of cells to replace apoptotic cells [157]. Several senescence-associated signaling molecules contribute to altered cellular proliferation and apoptosis in COPD. An inverse correlation was found between p16 INK4a expression and PCNA expression in alveolar epithelial cells and vascular endothelial cells, indicating that alveolar cell senescence is associated with a decrease in cellular proliferation and regeneration [71]. The expression of p16 is known to increase in aging cells and reduces the proliferation of stem cells [147]. Levels of the tumor-suppressor protein p53 are elevated in patients with emphysema who smoke [158], and p53 activation by CS induces endothelial cell apoptosis, which is inhibited by upstream p53 inhibition by macrophage migration inhibitory factor (MIF) [159].

## 7. Therapeutic Implications

There is increasing interest in the resolution of abundant senescence as a potential therapeutic approach in COPD. Here we mention some promising therapeutic approaches other than possible mRNA targets, sirtuin-activating drugs, and next-generation antioxidants well described by other reviews [160]. See Table 2 for a summary of therapeutic options.

### 7.1. Senolytics

Senolytic agents, compounds that facilitate the elimination of senescent cells, have received considerable attention lately as a potential treatment for COPD [160]. However, the investigation of these agents is limited by the lack of universal markers of senescence. A better understanding of pathways that induce and reinforce senescence in COPD may allow us to discover possible biomarkers that could serve as targets for these senolytic therapies [19,160,161]. The current proposed senolytic agents are compounds that activate sirtuins, PTEN, AMPK, or NRF2 or inhibit PI3K, mTOR, BCL-2/XL, FOXO4, and SASPs. Kaempferol and apigenin are also suggested senolytic agents.

Liu, et al. reported that 25(OH)D1alpha hydroxylase knockout mice, when compared with wild-type mice, had more DNA damage, ROS production, inflammatory infiltration of the colon, and production of inflammatory cytokines related to SASP [162]. 1,25-dihydroxyvitamin D3[1α,25(OH)2D3] has antiaging effects by upregulating nuclear factor (erythroid-derived 2) -like 2 (Nrf2), reducing ROS, decreasing DNA damage, reducing p16/Rb and p53/p21 signaling, increasing cell proliferation, and reducing cellular senescence and the SASP [163]. Supplementation with exogenous 1,25(OH)2D3 or with combined calcium/phosphate and the antioxidant N-acetyl-l-cysteine prolonged their average lifespan to more than 16 months and nearly 14 months, respectively [163].

### 7.2. Metformin

Metformin, a biguanide class and anti-diabetic drug, may prove beneficial for the treatment of COPD. Metformin reduces cardiovascular mortality, all-cause mortality, and cardiovascular events in type 2 diabetic patients with coronary artery disease [164]. Interestingly, a recent retrospective study demonstrated that metformin treatment for 2-years improved survival rates in COPD patients with type 2 diabetes [165]. Equally, reduced mortality was observed in patients with chronic lower respiratory diseases treated with metformin [166]. In an unmatched cohort study in Taiwan, type 2 diabetic patients who used metformin were less likely to develop COPD, with a hazard ratio of 0.56 (95% CI 0.537–0.584) [167]. In a prospective open-label trial of patients with moderate and severe COPD who also had type 2 diabetes, the use of metformin showed improvement in symptoms compared to baseline as measured by the St George’s Respiratory Questionnaire and transitional dyspnea index scores [168]. Metformin is also used to treat severe COPD exacerbations [169]. Metformin reduces the frequency of lung infections due to *Staphylococcus aureus* [170], *Pseudomonas aeruginosa* [171], and *Legionella* pneumonia [172] by modifying glucose flow across the respiratory epithelium. Whether these effects on exacerbations are senescence-based is unknown. Recently, it was suggested that activation of AMPK by metformin could reduce abnormal inflammatory responses in mice with elastase-induced emphysema, as well as cellular senescence [173]. Metformin prevented CSE-induced HBEC senescence and mitochondrial accumulation due to increasing DEPTOR expression [78].

### 7.3. Rapamycin

Cell senescence in COPD is linked to mTOR activation. Rapamycin binds to mTORs immunophilin FK-binding protein (FKBP12), and the Rapamycin FKBP12 complex then interacts with mTOR to inhibit its function [174]. Inhibition of mTOR with rapamycin prevented cell senescence and inhibited the proinflammatory SASP in mice and in lung vascular cells or alveolar epithelial cells [44]. Rapamycin also reduces inflammatory cells in BALF and decreases mean linear intercepts, destructive index, and mean alveolar airspace area in CS-exposed mice [175].

### 7.4. AMP-Activated Protein Kinase (AMPK) Activators

The AMPK activator, 5-aminoimidazole-4-carboxamide ribonucleotide (AICAR), is an analog of adenosine monophosphate that is capable of stimulating AMP-dependent AMPK activity, reduces CSE-induced IL6 and IL8 in HBECs (cell line BEAS-2B) and SAECs and elastase-induced emphysema in mice [173]. AICAR treatment also reduced the expression of p16, p21, and p66shc but augmented klotho gene expression in both BEAS-2B and SAECs treated with CSE, indicating the role of AMP as a therapeutic target in both inflammatory and senescent pathways [173].

### 7.5. Mitogen-Activated Protein Kinase (MAPK) Inhibitors

p38 MAP kinases are a family of four serine/threonine kinases activated by cytokines and cellular-induced stress. p38 MAPK is a key mediator of the SASP through regulation of NF-κB activity and stabilization of SASP effector mRNA in senescent fibroblasts [176,177]. Hongo, et al. reported the potential usefulness of a p38 MAPK inhibitor for the prevention of cellular senescence in cultivated human corneal endothelial cells [178]. p38 MAPK activation is increased in small airways of COPD patients and implicated in the pathogenesis of COPD [179]. Acumapimod, an oral p38 inhibitor currently undergoing clinical trials in severe acute COPD exacerbations, showed an improvement in lung function (forced expiratory volume in 1 s; FEV_1_) when compared to a placebo [180]. Doramapimod, a highly potent inhibitor of p38 MAPK, may also be effective in ameliorating inflammatory conditions in older populations [181]. Furthermore, Dabrafenib and Trametinib were approved by the FDA in 2018 for the treatment of melanoma and are used together to target different aspects of the MAPK pathway. Dabrafenib has anti-inflammatory properties, inhibiting hyperpermeability, CAM expression, and adhesion of leukocytes [182]. Therefore, MAPK inhibitors, similar to AMPK activators, may target inflammatory and senescent pathways.

### 7.6. B-Cell Lymphoma-2 (BCL-2) Inhibitors

BCL-2 inhibitors are a family of proteins which selectively inhibit the anti-apoptotic protein localized on the outer membrane of the mitochondria. Long-term retention of senescent cells can be attributed to increased expression of Bcl-family proteins, which can promote tissue damage through a SASP [183]. Chang, et al. showed that ABT263, an inhibitor of BCL-2 and BCL-xL, has the potential to selectively eliminate senescent cells through the induction of apoptosis [184]. Senolytic agents Venetoclax (ABT-199) and Navitoclax (ABT-263) induced apoptotic cell death in soft-tissue sarcomas [185]. Venetoclax was the first selective BCL-2 inhibitor to be approved for the treatment of chronic lymphocytic leukemia and acute myeloid leukemia [186]. Zeng, et al. suggested the involvement of BCL-2 in the pathogenesis of COPD [187]. BCL-2 inhibitors may have anti-neoplastic as well as anti-inflammatory and anti-aging properties.

### 7.7. Heat Shock Protein 90 (HSP90) Inhibitors

A recent study identified HSP90 inhibitors as possible senolytic agents by utilizing a library of compounds and identifying inhibitors of the HSP90 chaperone family as having significant senolytic activity in mouse and human cells [188]. HSP90, an ATP-dependent molecular chaperone involved in signal transduction, cellular transport, and protein destruction, is a therapeutic cancer target. Stroissnigg, et al. suggested that treatment of Ercc1−/Δ mice, a mouse model of a human progeroid syndrome, with the HSP90 inhibitor 17-DMAG significantly delayed the onset of several age-related symptoms [188]. In November 2021, clinical trials began with Gamitrinib, a mitochondrial HSP90 inhibitor with anti-TNF-receptor associated protein 1 (TRAP-1) and anti-neoplastic properties. Gamitrinib promotes the activation of cyclophilin D (CypD), mitochondrial permeability transition pore opening (MTPT), and the release of cytochrome c, which induces cell death [189]. Cells isolated from Trap1^−/−^ mice demonstrated both impaired cellular metabolic activity and impaired cellular division. Inhibiting HSP90 and TRAP-1 may potentially decrease the accelerated metabolic activity of senescent cells [189].

### 7.8. Eicosanoids

There are serval studies to suggest that eicosanoids could be utilized as a means of countering lung diseases and possibly senescence. Endogenous PGE2 suppresses inflammation via PGE receptor 4 (PGER4) activation. The PGER4 receptor agonist (ONO-AE1-329) modulated cytokine levels in asthma and COPD models [190]. Inhalation of exogenous PGE2 prevents bronchoconstriction provoked by aspirin [191,192]. Treatment with Celecoxib, a nonsteroidal anti-inflammatory COX2 inhibitor, inhibited interalveolar wall distance and pulmonary inflammation in the lungs of CS-treated rats [193]. Celecoxib prevents TNF-α-induced cellular senescence in human chondrocytes [194] but little is known about its role in regulating senescence in lung cells or tissue.

### 7.9. The Future of Senolytic Therapy

The following candidates for future senolytic therapies based on early-stage research were conducted in human cell lines in vitro: FOXO4-related peptides [195], previous mentioned BCL-2 inhibitors [196,197], USP7 inhibitors [198], Quercetin plus Dasatinib [199], Fisetin, [200,201] Piperlonguimine [200,202], BIRC5 gene knockout [203], GLS1 inhibitors [204], procyanidin C1 [205], and EF-24 [196]. Medications or potential therapeutic targets studied in either mice or xenograft models include src inhibitors/dasatinib [206], Navitoclax [207], senescence-specific killing compound 1 (SSK1)/gemcitabine [208], and anti-glycoprotein nonmetastatic melanoma protein B (anti-GPNMD) [209]. Equally, activation of nuclear factor-E2-related factor 2 (NRF2) through sirtuin signaling can alleviate oxidative stress by suppressing cellular senescence [210]. NRF2 signaling wanes during the aging process while senescent responses increase [211] and melatonin can enhance NRF2 responses to suppress senescence [212]. Melatonin is an antioxidant hormone produced primarily by the pineal gland and melatonin can also inhibit the p53-mediated senescence pathway through the elimination of ROS or p53 deacetylation induced by upregulation of SIRT1 expression [213]. Below, we highlight other possible candidates for future senolytic therapy.

#### 7.9.1. 25-Hydroxycholesterol (25HC)

25HC, an endogenous metabolite of cholesterol synthesis, represents a potential new class of senolytics. Limbad, et al. utilized single-cell RNA sequencing to identify *CRYAB,* a small heat shock protein, and *HMOX1* (heme oxygenase 1) as robust senescence-induced genes and senolytic targets. They further characterized 25HC, which interferes with CRYAB aggregation, and reported a decrease in fibro-adipogenic progenitor (FAP) and satellite cell (SC) concentrations in the presence of 25HC [214]. These two cell types are linked to muscle stem cell dysfunction in aged skeletal muscles [215]. 25HC targets CRYAB in many cell types, including the lung, and is localized in alveolar macrophages and pneumocytes of COPD patients [216].

#### 7.9.2. Azithromycin and Roxithromycin

Ozvari, et al. treated human MRC-5 and BJ fibroblast cell lines with a DNA-damaging agent and measured protein content as a determinant of cell viability. Treatment of human fibroblasts with azithromycin and roxithromycin induced aerobic glycolysis and autophagy, but their effects on mitochondrial oxygen consumption rates varied [202]. These glycolysis and autophagy changes may explain the potential senolytic activity of these macrolide antibiotics. Azithromycin is widely studied for the treatment of patients with COPD exacerbations; long-term administration suppresses inflammatory cytokine release, increases macrophage phagocytosis, and induces anti-inflammatory cytokine expression [217].

#### 7.9.3. Cardiac Glycosides

Cells undergoing oncogene-induced senescence (OIS), a sustained antiproliferative response due to an oncogene mutation or the inactivation of the tumor-suppressor gene [218], display cellular alterations in electrolytic chemical composition. Guerrero, et al. compared normal and senescent intracellular concentrations of sodium, calcium, and potassium using a fluorescent probe and found that senescent cells contained an increased number of these cations [219]. Treatment of these cells with Ouabain, an inhibitor of the Na^+^, K^+^-ATPase, induced a subset of pro-apoptotic BCL-2 proteins, activated JNK, GSK3-β, and p38 in senescent cells, and showed increased senolytic activity [219]. Moreover, Triana-Martinez and colleagues showed that the senolytic effect of cardiac glycosides was effective in the elimination of senescent-induced lung fibroblasts [220]. Ouabain and digoxin exhibit senolytic effects on p16-expressing human airway epithelial cells [221,222] suggesting there may be a potential benefit of using these medications to target senescent cells and prevent airway inflammation in patients with COPD.

**Table 2 medicina-58-00817-t002:** Therapeutic options to target senescence.

Medication/Therapeutic Target	Mechanism of Action	FDA Approved or Experimental
**AICAR:** **5-aminoimidazole-4-** **carboxamide riboside**	AMPK activation; reduces the expression of p16, p21, and p66shc [173]; reduces IL6 and IL8 in HBECs [173].	Experimental; only used in several investigations in humans [223,224,225].
**Metformin**	AMPK activation reduces elastase-induced emphysema and senescence in mice [173]; modifies glucose flow across respiratory cells [170,171,172].	Approved in 1995 for diabetes; used for PCOS; shown to inhibit the SASP [226].
**Rapamycin**	mTOR inhibition [44]; reduction of inflammation and mean alveolar space [175].	Approved in 1999 as an immunosuppressant agent [227,228].
**Acumapimod** **Doramapimod** **Dabrafenib/Trametinib**	p38 MAPK inhibition; regulation of NF-κB, and stabilization of SASP effector mRNA [177,178]; inhibition of CAM expression and leukocyte adhesion [183].	Experimental. Experimental. Approved in 2018 for the treatment of melanoma [172].
**Navitoclax (ABT-263)** **Venetoclax (ABT-199)**	BCL-2 inhibition leads to the induction of apoptosis through a SASP [184,185].	Approved, both in 2016 and 2020, for chronic lymphocytic leukemia and acute myeloid leukemia [229]. Tests were conducted in human cell lines in vitro [201].
**Gamitrinib**	HSP90 inhibition with anti-TNF-receptor associated protein 1 properties; activation of cyclophilin D; release of cytochrome c [189,223].	Experimental and undergoing clinical trials. Promising therapeutic advantages for pulmonary hypertension and senescence [230].
**FOXO4-DRI**	Blocks interaction of FOXO4 and p53 and prevents apoptosis [231].	Experimental promising senolytic [232]. Tests were conducted in human cell lines in vitro and in mice models [231].
**25-Hydroxycholesterol**	Interferes with CRYAB aggregation and decreases FAP and SC concentration [233]; localized in alveolar macrophages and pneumocytes in COPD patients [216].	Experimental. Currently considered for COVID-19 treatment [234]; evidence regarding possible role use in COPD [235].
**Melatonin**	Melatonin prevents senescence by activating Nrf2 and inhibiting ER stress [212] and p53 deacetylation induced by upregulation of SIRT1 [213]	Experimental.
**Matrine**	PI3K inhibition; Chinese herbal medication led to a reduced number of senescent cells; decreased IGF1 and pAKT [236].	Experimental. Found to induce apoptosis in acute myeloid leukemia [236].
**Dasitinib** **+ Quercetin**	Src tyrosine kinase inhibition; PI3K pathway inhibition [206,233].	Approved in 2017 for the treatment of leukemia in pediatric patients. Quercetin shows anti-inflammatory potential [237].
**Eicosanoids and COX2 inhibitors**	Endogenous PGE2 suppressed inflammation via PGE Receptor 4 (PGER4) activation. The EP4 receptor agonist (ONO-AE1-329) modulated cytokine levels in asthma and COPD models [190]; Inhalation of exogenous PGE2 prevents bronchoconstriction provoked by aspirin [191,192]; Celecoxib inhibited interalveolar wall distance and pulmonary inflammation in the lungs of CS-treated rats [193].	Experimental. Celecoxib is typically used to treat mild to moderate pain and help relieve symptoms of arthritis.
**Kaempferol** **Apigenin**	NF-κB p65 inhibition via activation of the IRAK1/IκBα signaling pathway [238].	Experimental.
**Azithromycin** **Roxithromycin**	Macrolide antibiotics that induce aerobic glycolysis and autophagy [202]; suppression of cytokine release in COPD exacerbations [217].	Approved in 1987 and 2002; used for prevention and treatment of exacerbations in COPD [217]. Tests were conducted in human cell lines [202].
**Ouabain** **Digoxin**	Cardiac glycosides that inhibit the Na^+^, K^+^-ATPase, induce pro-apoptotic BCL-2 proteins and activate JNK, GSK3-β, and p38 in senescent cells [219].	Ouabain is not approved in the USA, France, or Germany, but digoxin was approved in 2002. Tests were conducted in human cell lines and mice models [221,222].
**Gemcitabine**	SSKI with potent cytotoxicity for aged cells through interaction with MAPK pathway [208].	Approved in 2011 for metastatic breast cancer treatment. Tests were conducted in mouse models [208].

AMPK: AMP-activated protein kinase; mTOR: mammalian target of rapamycin; MAPK: mitogen-activated protein kinase; NFκB: nuclear factor kappa B; SASP: senescence-associated secretory phenotype; CAM: cell adhesion molecule; BCL-2: B-cell lymphoma 2; HSP90: heat shock protein 90; Anti-TNF-receptor: anti-tumor necrosis factor receptor; FOXO4: forkhead box O4; CRYAB: crystallin alpha B; FAP: fibro/adipogenic progenitor; SC: satellite cell; COPD: chronic obstructive pulmonary disease; Nrf2: nuclear factor-erythroid factor-2 related factor 2; PI3K: phosphoinositide 3-kinase; IGF1: insulin-like growth factor 1; pAKT: phosphorylated serine/threonine kinase; IRAK1: interleukin 1 receptor-associated kinase 1; JNK: c-Jun N-terminal kinases; GSK3-β: glycogen synthase kinase 3; SSKI: senescence-specific killing compound.

## 8. Conclusions

Overall, there is mounting evidence to suggest that senescence could contribute to cells being resistant to apoptosis, exhibiting elevated inflammation, and reduced dead cell clearance, resulting in extensive tissue remodeling observed in COPD. Targeting senescent cells using senolytics to selectively remove senescent cells or modulate SASP using small molecules or antibodies represents a novel approach to countering COPD progression. Several treatments that may target cellular senescence are in development.

There are many pathways linked to driving senescence, including DNA damage responses (due and not related to telomere length), telomere activity regulated by telomere length, capping and inhibition, DNA methylation pathways, the p53 pathway, the p16 and p21 pathways, the SIRTs, Klotho signaling, IGF1/Akt signaling, SA-βgal activity, ROS signaling, ADAM17 signaling, mTOR signaling, autophagy, phosphorylated H2AX, p38 MAPK signaling, NAD+/poly-ADP ribose polymerases mediated DNA damage repair, degradation of the transcription factor Sp1, ROS signaling, NFκB signaling, p21 pathway, ER stress, JAK/STAT signaling, RAS/PI3K/AKT signaling, mitochondrial DNA damage, TCA cycle, and mitochondrial DAMPs. Many of these pathways are linked to COPD initial and progression. However, the question remains whether targeting senescence will reverse all or some of these altered pathways. Equally, few studies have investigated telomerase activity in CS-exposures or disease conditions [239]. Several commercially available kits to measure telomerase assays, such as the TRAPeze™ RT Telomerase Detection Kit from Millipore/Sigma, could be useful approaches to determining telomere dysfunction. Although there is substantial evidence to demonstrate that senescence is occurring in COPD, many studies report elevated SASP but without evidence that these readouts were due to senescence directly. It is difficult to rule out the influence of other pathways playing a role in the production, release, and signaling of these SASP-associated mediators. Finally, would targeting senescence in COPD also treat other comorbidities? To determine whether therapies directly impact senescence-driven COPD, better markers of senescence and disease progression are needed. It is also important to study senescence through each stage of COPD progression to identify the best treatment strategy to begin administering senolytics for therapy.

## Figures and Tables

**Figure 1 medicina-58-00817-f001:**
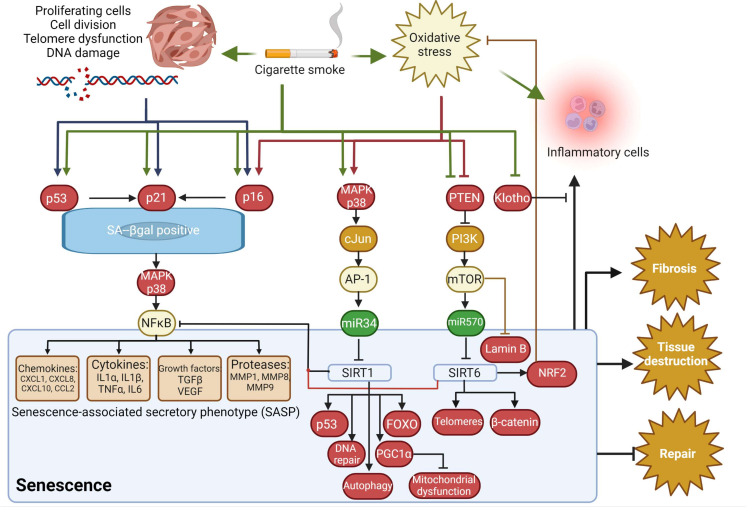
Possible mechanisms for senescence in COPD. Created with BioRender.com.

**Table 1 medicina-58-00817-t001:** Summary of the different types of senescence and readouts.

Senescence	SASPs	Known Pathways Involved
Replicative senescence	* Angiogenin, * bFGF, * CCL2, CCL3, CCL8, CCL16, * CCL20, * CCL26, * COX2, * CXCL1, CXCL2, CXCL3, * CXCR2, Fas, * FGF-7, * Fibronectin, * GM-CSF, * HGF, * ICAM-1, IFN-1, * IGFBP1, IGFBP2, IGFBP3, * IGFBP4, IGFBP5, IGFBP6, * IL1A, * IL1B, * IL6, IL7, * IL8, * IL11, * IL13, * IL15, * Leptin, * MIF, * MMP1, * MMP2, * MMP3, * MMP10, * Osteoprotegerin, * PAI-1, PAI-2, * PGE2, * PIGF, * SCF, * sgp130, sTNFRI, sTNFRII, * TGFβ, * TIMP2, * tPA, * TRAIL-R3, * uPA, * uPAR, and * WNT2	• DNA damage responses (due to telomere length) • * Telomere activity regulated by telomere length, capping and inhibition [16] • Derepression of the CDKN2A locus • DNA methylation pathways • * p53 pathway [24,25,26] • * p16 pathway [27] • * p21 pathway [16,28,29,30] • * SIRT1 [31,32] and * SIRT6 [31,33] signaling • * Klotho signaling [34] • * IGF1/Akt signaling [27] • * SA-βgal activity [24,26,35,36,37,38] • * ROS signaling [26,39,40] • Increased methylation of promoter of rDNA and reduced expression of 18S, 5.8S and 28S rRNA [41] • * S100A9 and TLR4 pathway [42]
Oncogene-induced Senescence (OIS)	* Angiogenin, * AREG, * A-SAA, * bFGF, * CCL1, * CCL2, CCL3, * CCL7, CCL8, * CCL13, CCL16, * CCL20, * CCL26, * COX2, * CXCL1, CXCL2, CXCL3, * CXCL5, * CXCL6, * CXCL7, * CXCL11, * CXCL12, * CXCL13, * CXCR2, * G-CSF, * GITR, * GMCSF, * HGF, * ICAM-1, IFN-1, * IFNγ, * IGFBP4, IGFBP6, * IGFBP7, * IL1A, * IL1B, * IL6, * IL6R, IL7, * IL8, * IL13, * LIF, * MIF, * MMP1, * MMP3, * MMP10, * Oncostatin M, * Osteoprotegerin, * PAI-1, * PGE-2, * PIGF, * sgp130, sTNFRI, * TNFRSF18, * t-PA, * TIMP1, * TIMP2, * uPAR, and * VEGF	• * p53/p21WAF1/CIP1 pathway [24,25,26] • * DNA damage responses (not due to telomere length) [28] • * ADAM17 signaling [43] mTOR signaling [44] • * Autophagy [45] • * Phosphorylated H2AX [29] • Nicotinamide phosphoribosyltransferase (NAMPT) activity • * ROS signaling [26,39,40] • * SA-βgal activity [24,26,35,36,37,38]
DNA-damage induced senescence	* Acrp30, * Amphiregulin, * Angiogenin, * bFGF, * BTC, * CCL1, * CCL2, CCL3, * CCL5, CCL8, * CCL13, CCL16, * CCL20, * CCL26, * CCL27, * CXCL1, CXCL2, CXCL3, * CXCL5, * CXCL6, * CXCL11, * EGFR, Fas, * FGF-7, * GDNF, * GM-CSF, * HGF, * ICAM-1, * IGFBP1, IGFBP2, IGFBP3, * IGFBP4, IGFBP5, IGFBP6, * IL1A, * IL1B, * IL6, * IL6R, IL7, * IL8, * IL11, * IL13, * IL15, * IL1R1, IL2Rα, * Leptin, * MIF, * MMP1, * MMP2, * MMP3, * MMP10, * MMP12, * MMP13, * MMP14, * MSP-a, * Oncostatin M, * Osteoprotegerin, * PDGF-BB, * PIGF, * SCF, * SDF-1, * sgp130, sTNFRI, sTNFRII, * TNFRSF18, * Thrombopoietin, * TIMP1, * TIMP2, * tPA, * TRAIL-R3, * uPA, * uPAR, and * VEGF	• * p16 pathway [27] • * p38 MAPK signaling [46] • NAD+/poly-ADP ribose polymerases mediated DNA damage repair • * p53 pathway [24,25,26] • DNA damage responses • Degradation of the transcription factor Sp1 • * ROS signaling [26,39,40] • * NFκB signaling [47] • * p21 pathway [16,28,29,30] • * Endoplasmic reticulum (ER) stress [48] • * SA-βgal activity [24,26,35,36,37,38]
Therapy-induced senescence	* AREG, * CXCL8, * IL1A, * IL-1B, * IL-6, * IL8 * MMP2, * MMP3, * PAI-1, * SPINK1, * t-PA, and WNT16B	• * p53 pathway [24,25,26] • JAK/STAT signaling • RAS/PI3K/AKT/mTOR signaling • * p16 pathway [27] • * p21 pathway [16,28,29,30] • * SA-βgal activity [24,26,35,36,37,38]
Mitochondrial dysfunctional-associated senescence	Lacks IL-1-dependent factors (* IL-1A, * IL-1B, * IL-6 and * IL8 are all reduced at the mRNA level), but includes * IL10, * CCL27, and * TNFα	• * ROS signaling [26,39,40] • * Telomere dysfunction [16] • * Mitochondrial DNA damage [49] • * Altered tricarboxylic acid (TCA) cycle [50] • * Reduced Parkin translocation [51] • * Cytoplasmic p53 accumulation [26] • Low NAD+/NADH ratios [52] • * Mitochondrial DAMPs [53] • Malic enzymes 1 and 2 • * Phosphorylated H2AX [29] • * SA-βgal activity [24,26,35,36,37,38]
Embryonic senescence	* AREG, * CCL2, * GM-CSF, * IL1A, * IL1B, * IL6, IL6R, * IL8, * ICAM1, * MIF, and * VEGF	• Shares many features to OIS • * p21 pathway [16,28,29,30] • * p15 pathway [24,25,26] • * TGFβ/SMAD and PI3K/FOXO pathways [54] • * Phosphorylated H2AX [29] • * SA-βgal activity [24,26,35,36,37,38]

* Denotes a link to playing a role in early emphysema. SASPs data based on [20,55,56,57,58,59,60]. Abbreviations: Acrp, catenin alpha like; AREG, amphiregulin; A-SAA, acute-phase serum amyloids; BTC, betacellulin; bFGF, basic fibroblast growth factor; CCL, chemokine (CC-motif) ligand; COX, cyclooxygenase; CXCL, chemokine (C-X-C motif) ligand; CXCR, C-X-C chemokine receptor; EGFR, epidermal growth factor receptor; FGF, fibroblast growth factors; G-CSF, granulocyte-colony stimulating factor; GDNF, glial cell-line derived neurotrophic factor; GM-CSF, granulocyte-macrophage colony-stimulating factor; HGF, hepatocyte growth factor; ICAM, intercellular adhesion molecule; IFN, interferon; IGFBP, insulin-like growth factor binding protein; IL, interleukin; LIF, leukemia inhibitory factor; MIF, macrophage migration inhibitory factor; MMP, matrix metalloproteinase; MSP, macrophage stimulating protein; PAI, plasminogen activator inhibitor; PDGF-BB, platelet-derived growth factor; PGE, prostaglandin; PIGF, placental growth factor; S100A9, S100 calcium-binding protein A9; SCF, stem cell factor; SDF, stromal cell-derived factor; sgp130, soluble glycoprotein 130; sTNFR, soluble tumor necrosis factor receptors; TGFβ, transforming growth factor beta; TIMP, tissue inhibitor of metalloproteinase; TNF, tumor necrosis factor; TNFRSF18, tumor necrosis factor receptor superfamily member 18; tPA, tissue plasminogen activator; TRAIL-R, tumor necrosis factor-related apoptosis-inducing ligand receptor; uPA, urokinase plasminogen activator; uPAR, urokinase plasminogen activator receptor; VEGF, vascular endothelial growth factor; WNT, Wingless and Int-1.

## Data Availability

Not applicable.

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
