# Peer review of "Senescence: Pathogenic Driver in Chronic Obstructive Pulmonary Disease"

_medicina, 2022, doi:10.3390/medicina58060817_

Round 1
Reviewer 1 Report
This is a very well written and comprehensive review of the current literature surrounding senescence in COPD. It is an emerging field with many outstanding questions. The authors have carefully described cellular senescence in general, as it pertains to COPD, the mechanisms of senescence, the pathologic consequences, and possible treatments. The structure and layout are clear but this reviewer would welcome the addition of at least 2 additional tables/figures to guide the audience.
Major comments
The review would benefit from at least 2 additional tables/figures to better illustrate the manuscript. Those new to the broader field of senescence outside of COPD would find the topic difficult to navigate given the vast numbers of players. Perhaps a simple table could be included early in the review to outline the different types of senescence (replicative, embryonic e.g. interdigital webs, extrinsic/disease-causing, e.g. cigarette smoke, genetic e.g. WRN gene mutations and oncogene?), and the known actors or pathways involved in each type, and whether is )? Given that emphysema is a clinical subtype of COPD, another useful table would be to list the genes so far implicated in early-onset emphysema. This smaller table could appear after section 4.1 with Figure 1 moving to next page.
It is unclear whether there is another type (or types) of telomere dysfunction apart from telomere shortening (section 4.1). If yes, how can this be measured (e.g. are telomerase activity assays possible?). Perhaps this could be clarified.
The SASP phenomenon seems poorly defined. How can we distinguish this from other signalling phenomena? The absence of a key hallmarks or signature unique to senescence seems a major obstacle – and an opportunity for researchers. Could a table be presented to describe the key outstanding questions in the field? Or at a minimum these should be clearly listed in the conclusion, which is too brief and provides an abrupt end to an excellent manuscript.
Lastly, the conclusion section is too brief and ends abruptly. The section undersells the importance of the review. At a minimum the conclusion section should describe the outstanding key questions for the field, and the next steps required.
Minor comments
· Line 48, insert space between “thesenescence”
· 227, please explain what Parkin is and does
· 234, delete extra space at end of line
· 235, insert space between “throughIL-6”
· 245, delete extra space after activity
· 276, explain what ZEB2 does
· 308, should this instead read “mTOR regulator”?
· 347, delete extra space after CK
· 356, delete comma
· 359, correct to “Alpha-1 antitrypsin Deficiency” and suggest AATD rather than AAT-D as this version is never used.
· 412, insert space “pathwaysincluding”
· 426, is the use of “immunoglobulins” correct here?
· 449, insert space in “suggestingthat”
· 456, should this be CS or CSE?
· 461, insert space in “andDNA”
· 470, change to “AATD”
· 483, delete extra space after “we”
· 528, insert space after full stop
· 546, delete extra space
· 554, insert space after comma following “properties”
· 567, insert space between “inhibitorsmay”
· 602, delete extra space between “and is”
· 615, please explain “oncogene-induced senescence (OIS)” as this is the first time it appears?
· Table 1. Gamitrinib line seems to contain an extra space after “trials”. Also the Matrine line has an extra space after full stop. Kaempferol line contains font of a different colour, see “signalling pathway”
· 635, different colour font evident for “interleukin 1 receptor associated kinase”.
Author Response
Reviewer #1
This is a very well written and comprehensive review of the current literature surrounding senescence in COPD. It is an emerging field with many outstanding questions. The authors have carefully described cellular senescence in general, as it pertains to COPD, the mechanisms of senescence, the pathologic consequences, and possible treatments. The structure and layout are clear but this reviewer would welcome the addition of at least 2 additional tables/figures to guide the audience.
Response: We thank the reviewer for their constructive feedback and we have tried to address all of your concerns/comments. We have added a new Table, now labeled Table 1. Rather than make two tables, we thought that we highlight SASPs and pathways of the different forms of senescence that are linked to early emphysema.
Major comments
The review would benefit from at least 2 additional tables/figures to better illustrate the manuscript. Those new to the broader field of senescence outside of COPD would find the topic difficult to navigate given the vast numbers of players. Perhaps a simple table could be included early in the review to outline the different types of senescence (replicative, embryonic e.g. interdigital webs, extrinsic/disease-causing, e.g. cigarette smoke, genetic e.g. WRN gene mutations and oncogene?), and the known actors or pathways involved in each type, and whether is )? Given that emphysema is a clinical subtype of COPD, another useful table would be to list the genes so far implicated in early-onset emphysema. This smaller table could appear after section 4.1 with Figure 1 moving to next page.
Response: We have added a new table, to summarize the types of senescence with their associated pathways and SASPs and pathways implicated in early-onset emphysema.
It is unclear whether there is another type (or types) of telomere dysfunction apart from telomere shortening (section 4.1). If yes, how can this be measured (e.g. are telomerase activity assays possible?). Perhaps this could be clarified.
Response: We have added two sentences regarding telomere capping to section 4.1 (lines 217-221) and also suggested the use of telomerase activity kits in the conclusion on lines 705-709.
The SASP phenomenon seems poorly defined. How can we distinguish this from other signalling phenomena? The absence of a key hallmarks or signature unique to senescence seems a major obstacle – and an opportunity for researchers. Could a table be presented to describe the key outstanding questions in the field? Or at a minimum these should be clearly listed in the conclusion, which is too brief and provides an abrupt end to an excellent manuscript.
Response: We agree with the reviewer that the current senescence field associates SASP “readouts” with senescence when they could easily be due to other signaling. Very few studies demonstrate causality. We have modified the conclusion to better reflect the outstanding questions in the field. Please see lines 709-713 on the version of the manuscript with highlighted changes.
Lastly, the conclusion section is too brief and ends abruptly. The section undersells the importance of the review. At a minimum the conclusion section should describe the outstanding key questions for the field, and the next steps required.
Response: We agree with the reviewer and have substantially changed the conclusion section. Please see lines 689-717.
Minor comments
- Line 48, insert space between “the senescence”
- 227, please explain what Parkin is and does
- 234, delete extra space at end of line
- 235, insert space between “throughIL-6”
- 245, delete extra space after activity
- 276, explain what ZEB2 does
- 308, should this instead read “mTOR regulator”?
- 347, delete extra space after CK
- 356, delete comma
- 359, correct to “Alpha-1 antitrypsin Deficiency” and suggest AATD rather than AAT-D as this version is never used.
- 412, insert space “pathwaysincluding”
- 426, is the use of “immunoglobulins” correct here?
- 449, insert space in “suggestingthat”
- 456, should this be CS or CSE?
- 461, insert space in “andDNA”
- 470, change to “AATD”
- 483, delete extra space after “we”
- 528, insert space after full stop
- 546, delete extra space
- 554, insert space after comma following “properties”
- 567, insert space between “inhibitorsmay”
- 602, delete extra space between “and is”
- 615, please explain “oncogene-induced senescence (OIS)” as this is the first time it appears?
- Table 1. Gamitrinib line seems to contain an extra space after “trials”. Also the Matrine line has an extra space after full stop. Kaempferol line contains font of a different colour, see “signalling pathway”
- 635, different colour font evident for “interleukin 1 receptor associated kinase”.
Response: We apologize for these errors and have made these corrections. We have made changes to address all the above comments, with the exception of:
- comment number 2 regarding Parkin, as we did state that “...Parkin translocation, a family of proteins which function as ubiquitin E3 protein-ligases (75), to damaged mitochondria and cytoplasmic p53 accumulation (47).” On lines 255-257
- should this instead read “mTOR regulator”? TSC1 is a negative regulator of mTOR
- 456, should this be CS or CSE? CSE was correct, i.e., cigarette smoke extract
Reviewer 2 Report
Dear Authors,
It was interesting to review the review " Senescence: Pathogenic driver in Chronic Obstructive Pulmonary Disease ".
I read this work with real interest. You reviewed the vast amount of literature and organized the knowledge of the topic they were interested in, in a thorough order. I was able to learn a lot by reading this work. However I would ask you for broadening your knowledge of a few aspects. By discussing the role of eicosanoids, you describe the pro-inflammatory role of PGE2 (line 331). Meanwhile, the facts about the anti-inflammatory importance of this substance are known. They also apply to COPD (1) or asthma (2). I suggest clarifying this knowledge. I am also curious if, according to above it could be used as one of therapeutic options.
Nrf2 inhibition described in table 1 should be more explained in main text. Additionally the role of melatonin should be described much more broadly.
References:
1. Birrell MA, Maher SA, Dekkak B, et al. Anti-inflammatory effects of PGE2 in the lung: role of the EP4 receptor subtype Thorax 2015;70:740-747
2. Mastalerz, L., Sanak, M., Gawlewicz-Mroczka, A., Gielicz, A., Cmiel, A., & Szczeklik, A. (2008). Prostaglandin E2 systemic production in patients with asthma with and without aspirin hypersensitivity. Thorax, 63(1), 27–34.
Author Response
Reviewer #2:
I read this work with real interest. You reviewed the vast amount of literature and organized the knowledge of the topic they were interested in, in a thorough order. I was able to learn a lot by reading this work. However I would ask you for broadening your knowledge of a few aspects. By discussing the role of eicosanoids, you describe the pro-inflammatory role of PGE2 (line 331). Meanwhile, the facts about the anti-inflammatory importance of this substance are known. They also apply to COPD (1) or asthma (2). I suggest clarifying this knowledge. I am also curious if, according to above it could be used as one of therapeutic options.
References:
- Birrell MA, Maher SA, Dekkak B, et al. Anti-inflammatory effects of PGE2 in the lung: role of the EP4 receptor subtype Thorax 2015;70:740-747
- Mastalerz, L., Sanak, M., Gawlewicz-Mroczka, A., Gielicz, A., Cmiel, A., & Szczeklik, A. (2008). Prostaglandin E2 systemic production in patients with asthma with and without aspirin hypersensitivity. Thorax, 63(1), 27–34.
Response: Thank you for your constructive comments. We have added a section on eicosanoids as suggested. These changes can be seen on lines 619-628 and in Table 2.
Nrf2 inhibition described in table 1 should be more explained in main text. Additionally the role of melatonin should be described much more broadly.
Response: We have added several new sentences on NRF2 on lines 637-644.
Reviewer 3 Report
1. a suggestion is o rewrite the text, clearly defining the feasible objectives to be met, confronted with the literature reviews, methods, results and discussion.
2. the abstract does not respect the IMRAD structure
3. the PRISMA guideliness need to be implemented in the article body, including the PRISMA flowchart for the included articles.
4. you did not specified the inclusion and the exclusion criteria, neither the period of the search.
5. you made a table for the treatment section.. but, taking into account that the title refers to pathogenic mechanisms in COPD, our suggestion is to synthetize the articles included in this section
6. in 368 line is Kumal or Kumar?
7. the reference 194 is Triana Martinez but in the article body (line 497) is Liu et al. !!!
8. there is no uniformity in the information presentation (treatment) - for example for rapamycin and AMPK activators you did not explained their mechanism of action.
9. the conclusion section does not respond to the article objective and title.
Author Response
Reviewer #3:
- a suggestion is o rewrite the text, clearly defining the feasible objectives to be met, confronted with the literature reviews, methods, results and discussion.
Response: We appreciate the meta-analysis approach of the reviewer but this format is not suited to this current manuscript as there are insufficient clinical trials performed to conduct a meta-analysis systemic review on this topic.
- the abstract does not respect the IMRAD structure
Response: This is true. But due to the nature of the review and the current literature, we have only performed the “I” (introduction) of the IMRAD structure. We are not conducting a study of studies. The IMRAD structure was not required in the authors' guidelines for a review paper.
- the PRISMA guideliness need to be implemented in the article body, including the PRISMA flowchart for the included articles.
Response: The PRISMA guidelines are not required in the authors' guidelines for a review paper.
- you did not specified the inclusion and the exclusion criteria, neither the period of the search.
Response: We did not limit the criteria for the inclusion of relevant literature in this review manuscript. We do not believe this to be a systematic review that required a meta-analysis approach.
- you made a table for the treatment section.. but, taking into account that the title refers to pathogenic mechanisms in COPD, our suggestion is to synthetize the articles included in this section
Response: It is not clear what this comment means. What does “synthetize the articles” mean?
- in 368 line is Kumal or Kumar?
Response: We have corrected this typo.
- the reference 194 is Triana Martinez but in the article body (line 497) is Liu et al. !!!
Response: We have corrected this referencing error.
- there is no uniformity in the information presentation (treatment) - for example for rapamycin and AMPK activators you did not explained their mechanism of action.
Response: We have added mechanisms of action for rapamycin and AICAR on lines 560-562 and 569-570.
- the conclusion section does not respond to the article objective and title.
Response: We agree with the reviewer and have substantially changed the conclusion section. Please see lines 689-717.
Round 2
Reviewer 3 Report
Congratulation for your work!